# Exploring the In Vitro and In Vivo Therapeutic Potential of BRAF and MEK Inhibitor Combination in NRAS-Mutated Melanoma

**DOI:** 10.3390/cancers15235521

**Published:** 2023-11-22

**Authors:** Heike Niessner, Anna Hüsch, Corinna Kosnopfel, Matthias Meinhardt, Dana Westphal, Friedegund Meier, Bastian Schilling, Tobias Sinnberg

**Affiliations:** 1Division of Dermatooncology, Department of Dermatology, University of Tuebingen, Liebermeisterstr. 25, 72076 Tuebingen, Germany; anna_huesch@web.de; 2Cluster of Excellence iFIT (EXC 2180) “Image Guided and Functionally Instructed Tumor Therapies”, 72076 Tuebingen, Germany; 3Department of Hematology, Oncology and Pneumology, University Hospital Muenster, 48149 Muenster, Germany; corinnaveronica.kosnopfel@ukmuenster.de; 4Department of Pathology, Medical Faculty and University Hospital Carl Gustav Carus, TU Dresden, 01307 Dresden, Germany; matthias.meinhard@uniklinikum-dresden.de; 5National Center for Tumor Diseases (NCT), Partner Site Dresden, 01307 Dresden, Germany; dana.westphal@uniklinikum-dresden.de (D.W.); friedegund.meier@uniklinikum-dresden.de (F.M.); 6Department of Dermatology, Carl Gustav Carus Medical Center, TU Dresden, 01307 Dresden, Germany; 7Center for Regenerative Therapies Dresden, TU Dresden, 01307 Dresden, Germany; 8Department of Dermatology, Venereology and Allergology, University Hospital Würzburg, Josef-Schneider-Str. 2, 97080 Würzburg, Germany; schilling_b@ukw.de; 9Department of Dermatology, Venereology and Allergology, Charité-Universitätsmedizin Berlin, Charitéplatz 1, 10117 Berlin, Germany

**Keywords:** malignant melanoma, cancer, encorafenib, binimetinib, BRAF inhibitor, NRAS-mutated, PDX (patient-derived xenograft) mouse model

## Abstract

**Simple Summary:**

This study explores the therapeutic effect of the combination of BRAF and MEK inhibitors (BRAFi + MEKi) on NRAS-mutated melanomas in vitro, in preclinical in vivo models using patient-derived xenografts (PDXs) and in an individual healing attempt of one patient. NRAS mutations are known to drive aggressive tumor growth and pose challenges for current treatments. This study aimed to assess the efficacy and safety of BRAFi + MEKi combination therapy in NRAS-mutant melanomas. This research seeks to provide critical insights into potential precision therapies for NRAS-mutant melanoma patients, ultimately advancing melanoma therapeutics and improving patient outcomes.

**Abstract:**

Introduction: Patients with NRAS-mutant metastatic melanoma often have an aggressive disease requiring a fast-acting, effective therapy. The MEK inhibitor binimetinib shows an overall response rate of 15% in patients with NRAS-mutant melanoma, providing a backbone for combination strategies. Our previous studies demonstrated that in NRAS-mutant melanoma, the antitumor activity of the MEK inhibitor binimetinib was significantly potentiated by the BRAFV600E/K inhibitor encorafenib through the induction of ER stress, leading to melanoma cell death by apoptotic mechanisms. Encorafenib combined with binimetinib was well tolerated in a phase III trial showing potent antitumor activity in BRAF-mutant melanoma, making a rapid evaluation in NRAS-mutant melanoma imminently feasible. These data provide a mechanistic rationale for the evaluation of binimetinib combined with encorafenib in preclinical and clinical studies on NRAS-mutant metastatic melanoma. Methods: The combination of BRAFi plus MEKi was tested in a monolayer culture of patient-derived cell lines and in corresponding patient-derived tissue slice cultures of NRAS-mutant melanoma. To investigate the treatment in vivo, NSG (NOD. Cg-*Prkdc*^scid^ *Il2rg*^tm1Wjl^/SzJ) mice were subcutaneously injected with three different BRAF wild-type melanoma models harboring oncogenic NRAS mutations and treated orally with encorafenib (6 mg/kg body weight, daily) with or without binimetinib (8 mg/kg body weight, twice daily). In parallel, an individual healing attempt was carried out by treating one patient with an NRAS-mutated tumor. Results: Encorafenib was able to enhance the inhibitory effect on cell growth of binimetinib only in the cell line SKMel147 in vitro. It failed to enhance the apoptotic effect found in two other NRAS-mutated cell lines. Encorafenib led to a hyperactivation of ERK which could be reduced with the combinational treatment. In two of the three patient-derived tissue slice culture models of NRAS-mutant melanomas, a slight tendency of a combinatorial effect was seen which was not significant. Encorafenib showed a slight induction of the ER stress genes *ATF4*, *CHOP*, and *NUPR1*. The combinational treatment was able to enhance this effect, but not significantly. In the mouse model, the combination therapy of encorafenib with binimetinib resulted in reduced tumor growth compared to the control and encorafenib groups; however, the best effect in terms of tumor growth inhibition was measured in the binimetinib therapy group. The therapy showed no effect in an individual healing attempt for a patient suffering from metastatic, therapy-refractory NRAS-mutated melanoma. Conclusion: In in vitro and ex vivo settings, the combination therapy was observed to elicit a response; however, it did not amplify the efficacy observed with binimetinib alone, whereas in a patient, the combinational treatment remained ineffective. The preclinical in vivo data showed no increased combinatorial effect. However, the in vivo effect of binimetinib as monotherapy was unexpectedly high in the tested regimen. Nevertheless, binimetinib proved to be advantageous in the treatment of melanoma in vivo and led to high rates of apoptosis in vitro; hence, it still seems to be a good base for combination with other substances in the treatment of patients with NRAS-mutant melanoma.

## 1. Introduction

In the field of oncology, significant advancements have been made in recent years in the treatment of melanoma, a type of skin cancer that originates in melanocytes, the cells responsible for producing melanin. Until 2010, therapeutic approaches known as chemotherapeutics had limited success in extending the overall survival (OS) of patients with metastatic melanoma [1]. The breakthrough came in 2011 with the introduction of the CTLA-4-specific antibody ipilimumab, which marked a significant milestone in improving patient survival [2]. Subsequently, in 2014, the FDA approved antiprogrammed cell death protein 1 (PD1) checkpoint inhibitors for the treatment of metastatic melanoma [3,4,5], revolutionizing disease management options. For patients with unresectable advanced stages, targeting PD1 has proven highly beneficial. PD1 inhibitors block cancer immunoinhibitory signaling, enabling melanoma patients to mount a robust antitumor response [6]. Depending on the design of clinical studies, treatment with the PD1 inhibitor nivolumab has led to sustained tumor regression in numerous patients [3,7,8]. 

Another crucial breakthrough has been the development of targeted therapies that specifically address the BRAFV600 mutation [9] driving the growth and spread of melanoma cells [10,11,12]. For patients with melanoma harboring NRAS mutations, a particular genetic melanoma subtype, treatment options with regard to targeted therapy are still very limited. A trial with the MEK inhibitor binimetinib alone did not yield the hoped-for results [13,14,15]. Limited treatment options are available for patients with NRAS-mutated melanoma, mainly consisting of participation in clinical trials like TIL therapy [16] or fecal microbiota transplantation [17,18], as well as chemotherapy. Consequently, therapy resistance poses a significant clinical challenge for as many as half of the patients undergoing immunotherapy. Furthermore, the tumors of patients with BRAF-WT melanomas are characterized by aggressive cell biology and an unfavorable prognosis [19,20].

The MAPK pathway has many interactions with other signaling pathways, and an emerging focus is its connection to endoplasmic reticulum (ER) stress and possible impacts on cell survival. It has been observed by several groups that BRAF inhibitors such as vemurafenib and encorafenib not only induce an inhibition of the MAPK pathway but also play a role in the induction of ER stress. However, the outcome and mechanisms of ER stress are under debate. Some data suggest that it is a potential target for overcoming BRAF inhibitor resistance by inhibiting its induction of autophagy. In contrast, other studies were able to prove the role of ER stress in the induction of apoptosis by BRAF inhibitors as a strategy to similarly treat NRAS-mutant melanoma [21,22,23,24].

ER stress is crucial for maintaining normal cell function and for determining cell fate. It can be triggered by several sensors and plays a role in the unfolded protein response as well as in autophagy induction. On the other hand, it can also switch from a cell maintenance program to a proapoptotic program when cells are irreversibly damaged [25]. In the context of cancer therapy, a possible mechanism of apoptosis induction by combined BRAF and MEK inhibitors treatments was proposed by our group [21]. BRAFV600E/K inhibitors such as encorafenib have been shown to activate the MAPK pathway in NRAS-mutant melanoma cells by increasing phosphorylated ERK (pERK). This also induces ER stress in several ways. One is the inhibition of the chaperone GRP78, which reduces its binding to PKR-like endoplasmic reticulum kinase (PERK). PERK phosphorylates the α-subunit of eukaryotic initiation factor 2 (eIF2α) and thereby enhances the downstream activation of ER stress-related factors such as ATF4, CHOP, and p8 (*NUPR1*). Under normal conditions, PERK is inhibited through the binding of GRP78. The upregulation of ATF4, CHOP, and p8 (*NUPR1*) results in the stimulation of the proapoptotic protein PUMA. However, the detailed mechanism behind the regulation of PERK by GRP78 is not completely understood. MEK inhibitors such as binimetinib have been shown to enhance the expression of the proapoptotic protein BIM. Altogether, these mechanisms result in the activation of the mitochondrial apoptotic pathway [21,22].

As previously mentioned, therapy options for patients with NRAS-mutated melanoma are still very limited and the disease is often associated with aggressive and fast tumor progression. Hence, there is a need for new therapies that work fast and effectively [26,27,28,29]. In BRAF-mutated melanoma, the combination of encorafenib and binimetinib showed outstanding response rates and overall survival. The overall response to binimetinib monotherapy in NRAS-mutated melanoma was 15%, making combination treatment a promising strategy for the treatment of NRAS-mutated melanoma [13]. Although BRAF inhibitors alone have been shown to induce metastasis in RAS-mutant melanomas by increasing ERK phosphorylation due to the so-called paradoxical activation of wild-type BRAF, a recent study was able to demonstrate that this effect could be reversed by a combination with an MEK inhibitor [21,30]. Coupled with the findings that BRAF inhibitors might induce ER stress in NRAS-mutant melanoma, leading to an amplification of MEK inhibitor activity, the combination of binimetinib and encorafenib might shift the balance towards better responses. The aim of this research was to further investigate the usefulness of combination therapy with encorafenib and binimetinib on NRAS-mutant melanoma models, including patient-derived xenografts and one individual healing attempt, and to address key considerations for its potential application in patients.

## 2. Materials and Methods

### 2.1. Isolation and Culture of Human Cells

Experiments were performed in accordance with the Declaration of Helsinki. The melanoma cell line SKMel147 (NRAS^Q61R^) was kindly provided by M. Soengas (Melanoma Laboratory, Molecular Pathology Programme, Centro Nacional de Investigaciones Oncológicas (Spanish National Cancer Research Centre, Madrid, Spain)). Mycoplasma infection in the cells was regularly checked using a Venor GeM Classic Mycoplasma Detection Kit (Minerva Biolabs, Berlin, Germany). The cells were cultured in RPMI 1640 medium supplemented with 10% fetal calf serum (FCS), 1% penicillin–streptomycin at 37 °C with 5% CO_2_ and 95% humidity. Patient-derived xenograft cell lines were isolated from melanoma tissue, which was cut into small pieces, digested in HBBS (*w*/*o* Ca^2+^ and Mg^2+^) with 0.05% collagenase, 0.1% hyaluronidase, and 0.15% dispase at 37 °C for 1 h, and filtered through a cell strainer (100 μm mesh). The single cells were taken into culture using an RPMI1640 medium supplemented with 10% FCS and 1% penicillin–streptomycin at 37 °C with 5% CO_2_ and 95% humidity. For cryopreservation, cell pellets were resuspended in Biofreeze medium (Biochrom/Merck, Berlin, Germany) and 1 mL of the cell suspension per cryotube was frozen for short-term storage at −80 °C and for long-term storage in liquid nitrogen.

### 2.2. Signaling Pathway Inhibitors and Treatments

Thapsigargin, encorafenib, and binimetinib were purchased from Selleck Chemicals (Cologne, Germany) and dissolved in DMSO for in vitro experiments or prepared as described in Section 2.9 for in vivo experiments.

### 2.3. Viability Assay

The viability of melanoma cells was assessed using the 4-methylumbelliferyl heptanoate (MUH) assay. Briefly, 2.5 × 10^3^ cells were seeded into cavities of a 96-well plate. After 24 h, cells were treated in hexaplicate for 72 h with increasing concentrations of thapsigargin (up to 100 nM)/encorafenib (up to 10 μM) or binimetinib (up to 10 µM), either as single or combinational treatments. Prior to analysis, cells were washed with phosphate buffered saline (PBS) and subsequently incubated with 100 μg/mL 4-methylumbelliferyl heptanoate diluted in PBS (1:100 dilution of 10 mg/mL stock) for 1 h at 37 °C. In viable cells, 4-methylumbelliferyl heptanoate is hydrolyzed by intracellular esterases and lipases, producing the highly fluorescent 4-methylumbelliferone. The fluorescence (λ_ex_ 355 nm, λ_em_ 460 nm) was detected with a Tristar fluorescence microplate reader (Berthold Technologies, Bad Wildbad, Germany). The intensity of fluorescence is indicative of the number of viable cells, and based on that, the relative viable cell number remaining after treatment was calculated normalized to the respective controls.

### 2.4. Cell Cycle Analysis

For the cell cycle analysis, 2.5 × 10^5^ cells were seeded into cavities of a 6-well plate and incubated for 2 h. Subsequently, the cells were treated with the indicated concentrations and combinations of thapsigargin/encorafenib and binimetinib for 3 days. Treatment was carried out in triplicate, and DMSO (0.02%) was used as a solvent control. To analyze the cell cycle distribution, floating and adherent cells were harvested, permeabilized with 70% ice-cold ethanol overnight, washed with PBS twice and resuspended in PBS with 50 μg/mL propidium iodide (Sigma-Aldrich, Taufkirchen, Germany) and 100 μg/mL RNAse A (AppliChem, Darmstadt, Germany). After staining for 30 min in the dark, the distribution of the cells in the different cell cycle phases was detected with a BD^TM^LSR II flow cytometer (BD Biosciences, Heidelberg, Germany) using FACSDiva^TM^ software version 6.1.3 (BD Biosciences, Heidelberg, Germany).

### 2.5. RNA Isolation and cDNA Synthesis

After treatment with the indicated concentrations and combinations of encorafenib and binimetinib for 24 h, the melanoma cells were harvested, and total RNA was extracted from the cell pellets using the RNeasy Micro Kit (Qiagen, Hilden, Germany) according to the manufacturer’s protocol. Complementary DNA was synthesized using the Reverse-Transcriptase Kit (Thermo Fisher Scientific, Sindelfingen, Germany) with 500 ng of total RNA, 4 μL of 5× RT buffer, 0.5 μL of Maxima reverse transcriptase (200 U/mL), 1 μL of random hexamer primer (100 μM), dNTPs (10 mM), and RNAse-free water to a total volume of 20 μL. After a preincubation of RNA with water for 10 min at 70 °C, the master mix was added and incubated for 10 min at 25 °C, followed by 45 min at 50 °C, and a final heat inactivation step for 5 min at 85 °C.

### 2.6. Quantitative Reverse Transcription–Polymerase Chain Reaction

Quantitative reverse transcription–polymerase chain reaction (qRT-PCR) was performed in a 10 μL reaction volume with GoTaq PCR Master Mix (Promega, Walldorf, Germany) (SYBR green-based) according to the manufacturer’s instructions using a Light Cycler 96 (Roche LifeScience, Penzberg, Germany). The initial denaturation step was at 95 °C for 5 min, followed by 40 cycles with 10 s each for the denaturation step at 95 °C, annealing at 60 °C, and elongation at 72 °C. Primer sequences are listed in Table 1. The primer oligos were purchased from the company Biomers (Ulm, Germany).

### 2.7. Western Blot

After treatment with the indicated concentrations and combinations of encorafenib and binimetinib for 24 h, the melanoma cells were lysed directly in the dish for 30 min on ice with a buffer containing 10 mmol/L Tris pH 7.5, 0.5% Triton X-100, 5 mmol/L EDTA, 0.1 mmol/L phenylmethylsulfonyl fluoride, 10 mmol/L pepstatin A, 10 mmol/L leupeptin, 25 mmol/L aprotinin, 20 mmol/L sodium fluoride, 1 mmol/L pyrophosphate, and 1 mmol/L orthovanadate. Lysates were cleared by centrifugation at 13,000× *g* for 30 min, and 15 to 60 mg of protein was subjected to SDS-PAGE and transferred to polyvinylidene difluoride (PVDF) membranes. Proteins were detected with antibodies against AKT (#9272), pAKT (Ser473) (#4060), ERK (#9102), pERKThr202/Tyr204 (#4376), ATF4 (#11815), and β-actin (#4970) (Cell Signaling Technology, Leiden, The Netherlands) horseradish peroxidase-conjugated secondary antirabbit IgG antibodies (#7074) (Cell Signaling Technology, Leiden, The Netherlands), and membranes were exposed to X-ray film (Eastman Kodak, Rochester, New York, NY, USA).

### 2.8. Tumor Slice Culture

Patient tissue samples were expanded in a patient-derived xenograft (PDX) mouse model by implanting the digested tumor tissue subcutaneously, as described previously [31]. After expansion, tumors were excised and cut with a Leica Microtome VT1200S into 400 µm slices. The slices were cultured in RPMI 1640 medium supplemented with 10% FCS, 1% penicillin, and 1% streptomycin and treated for 4 days with BRAF and MEK inhibitors in quadruplicate. An alamar blue assay was performed to analyze the viability of the tumor slices. In brief, 1 mg/mL of alamar blue stock solution was prediluted in culture medium (1:10), and 10 μL of this solution was added to 100 μL of culture medium of each sample. After incubation for 1 h at 37 °C, the fluorescence of resofurin was measured in the fluorescence microplate reader (Berthold Technologies, Bad Wildbad, Germany) at ex540 nm/em640 nm. For normalization of the reporter signals, the background signal was subtracted. Detailed information on the samples used are listed in Table 2.

### 2.9. In Vivo Mouse Experiment

All animal experiments were approved and performed in compliance with both European Union and German law and approved by local authorities (Regierungspraesidium Tuebingen, HT8/18G). NSG (NOD. Cg-Prkdc^scid^ Il2rg^tm1Wjl^/SzJ) mice were taken from our already ongoing in-house breeding (and housed) at the animal care facility at the University of Tuebingen. NSG mice received a subcutaneous (s.c.) right-flank injection of 3 × 106 melanoma cells (SKMel147, PDX129 or PDX62.1) suspended in 50 μL sterile PBS (Sigma-Aldrich, Taufkirchen, Germany). Once the tumors were established (tumor volume 40–80 mm^3^), the animals were randomly assigned to four groups (10 animals per group), and treatment was initiated. All mice were treated orally with the BRAFi encorafenib (6 mg/kg body weight daily dissolved in 0.5% methylcellulose with 0.5% Tween 80 (Sigma-Aldrich)) and/or the MEKi binimetinib (8 mg/kg body weight twice daily dissolved in 0.5% methylcellulose with 0.5% Tween 80 (Sigma-Aldrich, Taufkirchen, Germany)). The control group received the equivalent volume of solvent (0.5% methylcellulose with 0.5% Tween 80 (Sigma-Aldrich, Taufkirchen, Germany)) as treatment for a maximum of 28 days. Mouse weights were determined every second day throughout the experiment. Tumors were measured every day (volume = (length × width × width/2)). After 28 days or after the tumors had reached the critical size of 1500 mm^3^, the animals were euthanized. Tissue specimens, body fluids, and tumors were excised, collected, and stored for further investigations.

### 2.10. Immunohistochemistry of Mouse Tumors

For the immunohistochemical analysis, mouse tumor tissue was fixed in 4% formalin, embedded in paraffin, and stained with hematoxylin and eosin (HE). Proteins were detected with antibodies against Ki67 #M7240 (DAKO Agilent, Santa Clara, CA, USA) and pERKThr202/Tyr204 #4376 (Cell Signaling Technology, Leiden, The Netherlands). Bound antibodies were detected using UltraView Universal Alkaline Phosphatase Red Detection Kits from Ventana (Tucson, AZ, USA).

### 2.11. Immunohistochemistry of Human Tumors

Biopsies were taken from the inguinal region of a melanoma patient before and during encorafenib and binimetinib treatment. The tumor tissue was fixed in 4% formalin, embedded in paraffin, cut into 1–3 µm serial sections and dried for 30 min at 70 °C. Deparaffinization and immunostaining were performed using the BenchMark XT automated stainer (Ventana Medical Systems Inc., Tucson, AZ, USA), followed by dehydration and mounting of the slides. Detailed information about antigen retrieval, primary antibody incubation, and detection is provided in Table 3.

Positive control tissue was stained alongside the samples. HE staining was performed with the Sakura Tissue Tek Prisma automated stainer using a 1:2 mix from MERCK (#109249) and SAV (FSTL-HL-2500-M-1). For the semiquantitative evaluation, the staining intensity and the percentage of positive tumor cells were determined by two independent examiners. An immune reactive score (IRS) combining the staining intensity and the percentage of positive tumor cells was calculated [32]. Representative images taken at 100× and 400× magnification were processed for white balance using Photoshop v20.0.9 (Adobe). In Table 4, a detailed scoring is provided.

### 2.12. Individual Healing Experiment

A patient who had been diagnosed with metastatic melanoma and had prior treatments, including PD-1-based immunotherapy, was treated with encorafenib and binimetinib according to the approved protocol for BRAF-mutated patients. The patient’s informed consent was obtained, and a detailed medical history was recorded to ensure eligibility for the experimental treatment. The patient was closely monitored throughout the individual healing experiment, with regular assessments and documentation of treatment-related adverse events.

### 2.13. Statistics

Data were statistically analyzed with GraphPad Prism version 8.4. For multiple-group comparisons, a one-way ANOVA with subsequent Kruskal–Wallis test was used for the *p*-value calculation and significance determination. *p*-values < 0.05 were considered statistically significant (*: *p* ≤ 0.05; **: *p* ≤ 0.01; ***: *p* ≤ 0.001; ****: *p* ≤ 0.0001). To assess the potential synergy of inhibitor combinations, we utilized CompuSyn software 1.0 from ComboSyn Inc. (www.combosyn.com) to calculate combination indices (CIs). CI values of 1 denote additive effects, whereas values below 1 denote synergistic effects, and values above 1 denote antagonistic effects [33].

## 3. Results

### 3.1. Effects of Thapsigargin/Encorafenib and/or Binimetinib on the Viability and Cell Cycle of Melanoma Cells

To first prove that an ER stress inducer can enhance the apoptotic effect of binimetinib, the MEK inhibitor was combined with the classical ER stress inducer thapsigargin. Binimetinib was used up to 1 µM and thapsigargin up to 100 nM. Indeed, we could show that combining the two agents led to a higher growth inhibition rate (Figure 1A, red curve) and to a higher induction of apoptosis shown in a cell cycle analysis (Figure 1C, red part of the bar). Since thapsigargin cannot be applied to patients, we decided to further analyze the combination of encorafenib (BRAFi) and binimetinib (MEKi), because it was shown before that BRAF inhibitors were able to induce ER stress [21,22].

To test the effects of binimetinib and/or encorafenib on NRAS-mutated cells (cell line SKMel147 and patient-derived xenograft cells PDX129 and PDX62.1), viability was measured. Binimetinib and encorafenib were used at a range from 0.039 to 10 µM. As expected, monotherapy with the BRAF inhibitor encorafenib did not show strong effects in this panel of NRAS-mutated cells. The cell line PDX62.1 showed a decrease in viability at 5 µM BRAFi (Figure 1G, green curve). The best responses upon binimetinib treatment were observed in the cell lines SKMel147 (Figure 1D, blue curve) and PDX62.1 (Figure 1J, blue curve). The effect of the combination could significantly reduce the viability in the cell line SKMel147 (Figure 1D, red curve and Figure 1E) and at the higher concentration in PDX129 (Figure 1G, red curve and Figure 1H). The combination index analysis (Figure 1B,E,H,K) showed a synergism (CI < 1) for the addition of thapsigargin but also for that of encorafenib to binimetinib in SKMel147 and to a lesser extent in PDX129 and PDX62.1 cells, especially at higher concentrations. Of note, antagonistic effects were revealed at low concentrations (CI > 1). The cell cycle analysis revealed that the effect of binimetinib on apoptosis induction, which could be observed in all three cell lines (Figure 1F,I,L, red part of the bar), could not be enhanced but was also not reduced by the combination. After the combinational treatment, 35.9% (with 0.1 µM binimetinib and 1 µM encorafenib) and 39.4% (with 1 µM binimetinib and 1 µM encorafenib) of SKMel147 cells were found in the subG1 fraction (Figure 1F). For PDX129, the combinational treatments led to 31.0% (with 0.1 µM binimetinib and 1 µM encorafenib) and 70.8% (with 1 µM binimetinib and 1 µM encorafenib) of cells in the subG1 fraction (Figure 1I), whereas in PDX62.1, the combinational treatments caused 45.6% (with 0.1 µM binimetinib and 1 µM encorafenib) and 74.9% (with 1 µM binimetinib and 1 µM encorafenib) of subG1 cells (Figure 1L).

### 3.2. Effects of Binimetinib and Encorafenib on ER Stress Gene mRNA and Protein Expression

To investigate the effect of binimetinib and/or encorafenib on the expression levels of the ER stress genes *ATF4*, *CHOP*, and *NUPR1*, quantitative RT-PCR and Western blotting were performed. In the cell line SKMel147, encorafenib alone was already able to upregulate the RNA expression of *ATF4* compared to the control (Figure 2A). The RNA levels of *CHOP* and *NUPR1* were only enhanced after combination therapy with binimetinib plus encorafenib (Figure 2B,C). At the protein level, ATF4 was also already enhanced after a single treatment with encorafenib and to the same level after a combinational treatment with binimetinib and encorafenib.

The protein levels of phosphorylated AKT (pAKT) were slightly decreased after treatment with encorafenib and after combination treatment (Figure 2D). The MEK inhibitor binimetinib was able to block the phosphorylation of ERK (pERK) (Figure 2D). As described previously [30], encorafenib increased ERK phosphorylation in NRAS-mutant melanoma cells. However, the MEKi binimetinib was able to counteract this paradoxical activation of the MAPK pathway, resulting in a lower level of ERK phosphorylation after the combination treatment compared to the sample treated with the BRAFi encorafenib alone (Figure 2D).

### 3.3. Effects of Binimetinib and Encorafenib on the Ex Vivo Tumor Viability of NRAS-Mutated Tumors

To test whether binimetinib plus encorafenib also affected the survival of NRAS-mutant melanoma cells in a more physiological context, tumor tissue excised from two patients with NRAS-mutant melanoma and the established cell line SKMel147 were expanded in a mouse PDX model, and finally, the tumor was removed and prepared for tumor slice cultures. After treatment of the slices with MEKi and BRAFi, the tumor cells in these tissue slice cultures displayed a reduced viability compared to the untreated controls, especially in models PDX129 and PDX62.1 (Figure 3A–C), whereas there was no significant difference compared to the slices treated with the binimetinib monotreatment.

### 3.4. Case Report of Binimetinib and Encorafenib in a Patient with Advanced NRAS-Mutated Melanoma

In the clinic, a patient with advanced NRAS-mutant melanoma was treated as an individual healing attempt based on previous results [21]. This patient had been diagnosed with metastatic melanoma, and all relevant treatment options had been exploited, including immune checkpoint inhibition (ICI) with anti-PD1 and the combination of anti-CTLA4 plus anti-PD-1. Throughout all therapies, the participant showed massive progression. The patient’s informed consent was obtained in 08/2019, and a detailed medical history was recorded to ensure eligibility for the experimental treatment (Figure 4A). The patient was closely monitored throughout the individual healing experiment (Figure 4B), with regular assessments and documentation of treatment-related adverse events. The patient further progressed under the newly assessed treatment with binimetinib plus encorafenib; therefore, the therapy was stopped after 6 weeks in 09/2019.

To investigate the effects of encorafenib and binimetinib, we analyzed the patient’s tumor tissue taken before and during treatment. The biopsies were stained for the proliferation marker Ki67, the MAPK marker ERK/pERK and the survival marker AKT/pAKT. There was an increase in Ki67-positive cells (Figure 4C) as well as an upregulation of pERK and pAKT on-treatment compared to the before-treatment biopsy (Figure 4D), indicating an increased proliferative and survival activity of the tumor after treatment initiation. A detailed summary of the staining results is provided in Table 5.

### 3.5. Effects of Binimetinib and Encorafenib In Vivo

To assess the treatment efficacy of binimetinib and/or encorafenib in more detail, NSG mice received a subcutaneous (s.c.) right-flank injection of melanoma cells (SKMel147, PDX129, or PDX62.1). Once the tumors were established, the animals were randomly assigned to four groups (10 animals per group), and treatment was initiated (Figure 5A). All mice were treated orally with the BRAFi encorafenib (6 mg/kg body weight daily dissolved in 0.5% methylcellulose with 0.5% Tween 80 (Sigma-Aldrich)) and/or the MEKi binimetinib (8 mg/kg body weight twice daily solved in 0.5% methylcellulose with 0.5% Tween 80 (Sigma-Aldrich)). The control group received the equivalent volume of solvent (0.5% methylcellulose with 0.5% Tween 80 (Sigma-Aldrich)) for a maximum of 28 days. Tumors were measured every second day, and the volume was calculated as follows: volume = (length × width × width/2)). The animals were euthanized after 28 days of therapy or after the tumors had reached the critical size of 1500 mm^3^.

The tumor volumes monitored over time are shown In Figure 5B for SKMel147, in Figure 5D for PDX129, and in Figure 5F for PDX62.1. BRAF inhibition with encorafenib 6 mg/kg (green curves) did not change tumor growth compared to the sham-treated control mice (black curves), whereas the use of the MEK inhibitor binimetinib (8 mg/kg) significantly diminished tumor growth in all three models. MEK inhibition permanently inhibited tumor growth in the PDX129 and PDX62.1 models, whereas in the SKMel147 in vivo model, the tumors started to grow out after 10–12 days of treatment (blue curves). On the one hand, the combination therapy of encorafenib with binimetinib (red curves) resulted in significantly reduced tumor growth compared to the control and encorafenib groups; on the other hand, inhibition was significantly worse compared to binimetinib monotherapy. In all three models, treatment with the MEKi binimetinib alone resulted in the strongest tumor growth delay. However, the combination slowed tumor growth and progression when compared with the sham-treated mice.

The survival of the mice was evaluated by mouse health reports based on the following parameters: tumor size, health score, and weight development. Events occurred when the abortion criteria of the animal experiments were reached (in conjunction with the 3R principles for animal experiments). The mice treated with binimetinib survived throughout the 28 days of therapy in the PDX129 (Figure 5E) and PDX62.1 (Figure 5G) melanoma models, whereas three death events occurred before the end of the experiment in the SKMel147 (Figure 5C) model. However, the median survival was not reached in any of the models treated with binimetinib. For the control and encorafenib-treated mice, the median survival was approximately 13 days in the SKMel147 model, approximately 23 days in the PDX129 model, and was not reached in the PDX62.1 model. The combination therapy was almost as good as the binimetinib monotherapy in terms of survival, except for the SKMel147 model.

### 3.6. Immunohistochemical Evaluation of the PDX Tumors

For further analysis, immunohistochemical staining of the nuclear proliferation marker Ki67 was performed for the tumor tissue (Figure 6A for SKMel147, Figure 7A for PDX129 and Figure 8A for PDX62.1). Proliferating cells were stained red. An ImageJ-based Ki67 module for pathological application was used, and the percentage of proliferating cells at the center and margin of every individual biopsy was determined. The mean of all treatment groups was calculated for the different PDX models. The results of the PDX tumors are summarized in Figure 6B for SKMel147, in Figure 7B for PDX129, and in Figure 8B for PDX62.1. In the PDX129 model, treatment with binimetinib led to a significantly reduced proportion of proliferating cells at the tumor margin. This trend was also observed in the center of the corresponding tumors. Since only detectable tumor nodules could be evaluated, the effects on Ki-67 staining were not as strong as those observed in terms of tumor growth. This means that tumors that started to grow at the end of the experiment showed similar proliferation rates to the control tumors of sham-treated mice. This indicates the development of a resistance mechanism.

The tumor models were also stained for pERK to determine the MAPK activation level in the tumor specimens (Figure 6C for SKMel147, Figure 7C for PDX129, and Figure 8C for PDX62.1). For the evaluation of the pERK content of a tumor, all pictures within a tumor model were compared, and intensities ranging from zero (absent) to three (strong) were defined. This was performed separately for each individual tumor model since the tumors were not stained simultaneously. After intensity scores of 0–3 were defined for each tumor model, the images were divided into quadrants, and each quadrant was evaluated, resulting in a mean pERK score per biopsy. The average with standard deviation for every therapy group was determined per tumor model (in Figure 6D for SKMel147, in Figure 7D for PDX129, and in Figure 8D for PDX62.1). A reduction in pERK could be detected in the PDX129 model due to therapy with binimetinib. However, here, the combination also led to the reduced phosphorylation of ERK1/2 in the tumor margin. Similarly, significant results could be seen in the PDX62.1 model. This shows that the treatment groups containing binimetinib were effective in blocking MAPK signaling activity. No effect on pERK levels could be detected in the SKMel147 model. This is probably because the binimetinib tumors started to grow at the end of the 28-day therapy period.

Taken together, it becomes evident that the combination was not superior to monotreatment with binimetinib in its applied form. The final tumors also did not reveal a marked upregulation of ER stress-related genes. However, the combination also showed significant growth inhibition in all three in vivo models when compared with the sham-treated control groups. This translated into improved survival in terms of survival for PDX129 and PDX62.1. For the fast-growing melanoma model SKMel147, the combination performed worse in terms of survival when compared with the binimetinib treatment as a monotherapy.

## 4. Discussion

NRAS-mutant melanoma is usually accompanied by aggressive and rapid disease progression [19,20]. Hence, within the field of melanoma research, it remains a major objective to find effective therapy options in the treatment of patients with NRAS-mutant melanoma [34,35]. Currently, there is no specific therapy available to directly target NRAS mutations. As a result, MEK inhibition stands as the sole targeted therapeutic option for this particular melanoma subtype. Among the MEK inhibitors, binimetinib has emerged as a notably effective choice for treating NRAS-mutant melanoma. In phase 2, binimetinib showed activity in patients with NRAS-mutated melanoma as a first-targeted therapy [36]. In the phase 3 NEMO trial, binimetinib somewhat prolonged PFS and OS in patients with metastatic NRAS-mutated melanoma compared with dacarbacin [13]. However, the effect was small, highlighting the need for innovative combination therapies. Initial studies have highlighted its potential as a foundational component for innovative combination treatment strategies [28]. Our recent in vitro study unveiled a promising approach in which the BRAF inhibitor encorafenib, although being a V600E/K-specific inhibitor, had synergistic effects when combined with binimetinib in NRAS-mutated melanoma cells in terms of increased viability reduction and apoptosis induction. There, the addition of encorafenib complemented the proapoptotic effects of binimetinib by inducing ER stress that promoted the proapoptotic cascade including Bim and PUMA [13,21,36]. Therefore, the aim of the present study was to test the applicability of this combination in vivo using three xenograft models as well as to bolster the concept that BRAF inhibitors enhanced the antitumor efficacy of MEK inhibitors in NRAS-mutated melanoma. This study transitioned the combination therapy of binimetinib and encorafenib from the realm of preclinical in vitro testing to in vivo models. However, the findings indicated that further research in this domain is warranted. Across the three distinct PDX models, tumors treated with binimetinib monotherapy exhibited the most favorable responses. They displayed slower growth rates and improved overall survival, with survival rates remaining above 60 percent. On the other hand, the benefits of the combination treatment varied among the PDX models. Particularly in the SKMel 147 mice, tumors exhibited a slower growth compared to untreated or encorafenib-only treatment but developed more dynamically toward the end of the treatment period. This phenomenon is reminiscent of a more rapid development of resistance to MAPK pathway inhibition [37]. One explanation may be the paradoxical hyperactivation of wild-type BRAF when V600E/K-specific inhibitors are used [38,39]. This can be interpreted as an off-target effect of the BRAFi. It was revealed that wild-type BRAF bound to the inhibitor dimerizes with CRAF, leading to a paradoxical hyperactivation of the pathway [40]. This effect can be largely suppressed with MEK inhibitors [21,41,42,43]. However, the underlying mechanism could accelerate the reactivation of the MAPK pathway and thus the development of resistance in our models. Of note, the single treatment of mice with encorafenib did not result in a faster tumor growth compared with the untreated tumors in all three models, suggesting that the effect of the paradoxical activation on resistance development may outweigh the effect on tumor growth. In line with this, no significant increase in phosphorylated ERK1/2 was detected in the BRAFi-only treated tumors. However, although statistically not significant, a trend towards higher phospho-ERK1/2 levels was observed in the combination group versus the binimetinib group. This, again only by trend, was associated with a high Ki67 positivity and thereby high proliferation rates. Similarly, it is also worth noting that the in vivo outcome of the case report summarized in Figure 4 did not consistently align with the (until then promising) in vitro data. In this case, however, the patient had undergone multiple treatment regimen and had not responded to immunotherapy (neither monotherapy nor combination treatment). Biopsies from the patient on combination therapy also showed signs of resistance and hyperactivation compared with baseline, as evidenced by increased phosho-ERK1/2 and phospho-AKT levels and intense Ki67 staining. Although there were signs of synergism in the in vitro data, lower concentrations of encorafenib plus binimetinib also showed a high combination index and thereby an antagonistic behavior. We assume that higher levels of BRAFi are required to induce ER stress, and this may be the result of off-target effects that also affect the endoplasmic reticulum although no off-targets other than RAF isoforms are known for encorafenib [44]. It seems likely that the applied dosages in the mice did not translate into the tissue concentrations of inhibitors required for a synergistic effect.

Clinically, the development of resistance to BRAF inhibitors often involves the selection of pre-existing subpopulations of cancer cells with alternative genetic alterations [45,46]. NRAS mutations may already exist as subclones within the tumor before treatment. As BRAF inhibition exerts a selective pressure on BRAF-mutated cells, it may inadvertently favor the survival and expansion of NRAS-mutated cells, leading to their emergence as dominant clones. NRAS mutations activate the MAPK pathway independently of BRAF mutations [47]. When NRAS-mutated cells become dominant in the tumor, they can reactivate MAPK signaling, circumventing the inhibitory effects of BRAF inhibitors. This reactivation promotes cell survival and proliferation, contributing to disease progression. This can be a problem regarding the treatment of NRAS-mutated tumors with BRAF inhibitors, although combined with MEK inhibitors.

In our xenograft models, we experienced a surprisingly strong effect of the MEKi binimetinib in terms of tumor growth inhibition. While the primary goal of MEK inhibitors is indeed to directly suppress cancer cell growth by blocking aberrant MAPK signaling, emerging research has unveiled an intriguing secondary benefit: the induction of immunological effects within the tumor microenvironment. MEK inhibition can increase the infiltration of cytotoxic T lymphocytes (CTLs) into the tumor site. These CTLs are responsible for recognizing and eliminating cancer cells and also show a predictive capacity for the response to ICIs [48]. The presence of tumor-infiltrating lymphocytes is associated with improved clinical outcomes in various cancers, highlighting the potential immunological benefits of MEK inhibitors [49,50]. However, in a study where the MEK inhibitor cobimetinib was combined with the anti-PD-L1 checkpoint inhibitor atezolizumab for BRAF wild-type patients, there was no improved progression-free survival compared to monotherapy with pembrolizumab (anti-PD-1) alone [51].

The combination of ER stress induction and MAPK pathway inhibition represents a promising approach in cancer therapy. By simultaneously exploiting the vulnerabilities of cancer cells under ER stress and targeting the dysregulated MAPK pathway, this strategy holds the potential to enhance antitumor effects and overcome resistance mechanisms. Continued research and clinical trials are essential to further elucidate the optimal combination strategies [52,53], dosages, and patient populations for this innovative therapeutic approach. Ultimately, the convergence of ER stress and MAPK pathway inhibition may offer new hope for patients with cancer, paving the way for more effective and personalized treatment strategies. In addition, other combinatorial approaches using MEKi such as the addition of panRAF inhibitors [54,55,56], ERK inhibitors [57], BET inhibitors [58], or PI3K inhibitors [59] could be a step forward for having an effective therapeutic weapon in the fight against aggressive NRAS-mutated melanomas.

## 5. Conclusions

BRAF and MEK inhibitors do not appear to be a promising therapeutic regimen for the treatment of melanoma patients with NRAS-mutated metastatic melanoma when applied at the currently used doses for the treatment of BRAFV600E/K melanoma. This is due to the risk of re- or hyperactivation of the MAPK pathway by inhibitors for the mutated BRAF in BRAF wild-type settings with subsequent disease progression. However, in vitro data suggest that combinations such as panRAFi and MEKi can induce ER stress, causing melanoma cells to initiate programmed cell death. As this effect has yet to be confirmed in vivo, further research on such novel drug combinations is needed to develop an effective therapy against NRAS-mutated melanomas.

## Figures and Tables

**Figure 1 cancers-15-05521-f001:**
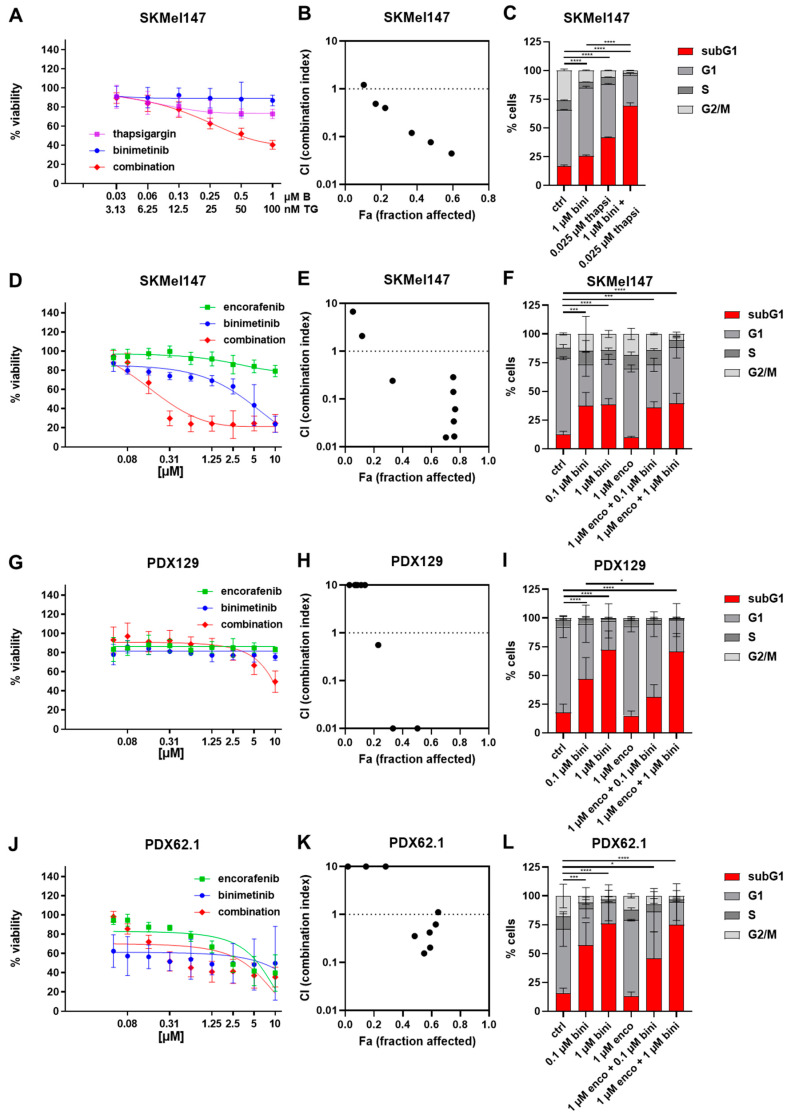
Effects of the combination of thapsigargin/encorafenib and/or binimetinib on the viability and cell cycle of melanoma cells. (**A**) SKMel147 cells were treated with thapsigargin (=TG) (up to 100 nM), binimetinib (=B) (up to 1 µM), and the combination for 72 h. (**D**) SKMel147, (**G**) PDX129, and (**J**) PDX62.1 cells were treated with binimetinib (up to 10 µM), encorafenib (up to 10 µM), and the combination for 72 h. Measured is the viability of treated cells compared to the untreated control. Shown are the mean values with the standard deviations (SDs) of three independent experiments, each measured in quadruplicate and normalized to the untreated control. Both drugs were used in equimolar ranges, so the *x*-axis labeling applies to both drugs. (**B**,**E**,**H**,**K**) show the synergism analysis of (**B**) SKMel147 cells treated with thapsigargin, binimetinib, and the combination; and (**E**) SKMel147, (**H**) PDX129, and (**K**) PDX62.1 cells after treatment with binimetinib, encorafenib, and the combination for 72 h. (**C**) SKMel147 cells were treated with thapsigargin, binimetinib, and the combination; and (**F**) SKMel147, (**I**) PDX129, and (**L**) PDX62.1 cells were treated with binimetinib, encorafenib, and the combination. The cell cycle after PI staining was measured by flow cytometry. Shown are the mean values of each cell cycle fraction with the SDs of three independent experiments, each measured in triplicate. Only the subG1 fractions were statistically compared as indicated (*: *p* ≤ 0.05; ***: *p* ≤ 0.001; ****: *p* ≤ 0.0001).

**Figure 2 cancers-15-05521-f002:**
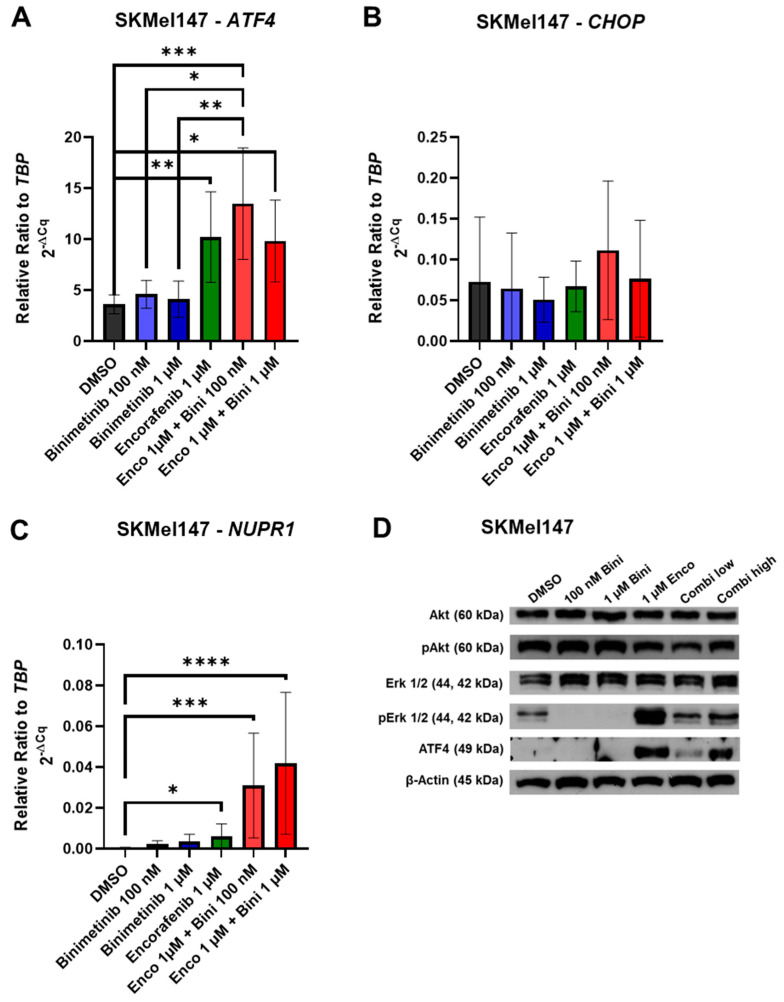
Effects of binimetinib and encorafenib on ER stress genes and MAPK and PI3K signaling. Quantitative real-time qPCR analysis of (**A**) *ATF4*, (**B**) *CHOP*, and (**C**) *NUPR1* in SKMel147 melanoma cells normalized to the reference gene TBP. Cells were treated for 72 h with 100 nM or 1 µM binimetinib, 1 µM encorafenib, and a combination of 1 µM encorafenib with either 100 nM or 1 µM binimetinib. The relative ratio was calculated with LightCycler^®^ 96 software version 6.1.3. Samples were measured in triplicate, and the values are the means ± SD of three replicates. (**D**) Phosphorylation status of ERK and AKT and the expression of ATF4 in SKMel147 cells. Western blot analysis of pERK/ERK, pAKT/AKT and ATF4 expression levels in NRAS-mutant SKMel147 cells. β-Actin was used as a reference protein, and the samples were loaded on a 10% SDS polyacrylamide gel. Cells were treated for 24 h with 100 nM or 1 µM binimetinib, 1 µM encorafenib, or a combination of 1 µM encorafenib with either 100 nM or 1 µM binimetinib. DMSO without treatment was used as a control sample (*: *p* ≤ 0.05; **: *p* ≤ 0.01; ***: *p* ≤ 0.001; ****: *p* ≤ 0.0001). The uncropped blots are shown in Appendix A.

**Figure 3 cancers-15-05521-f003:**
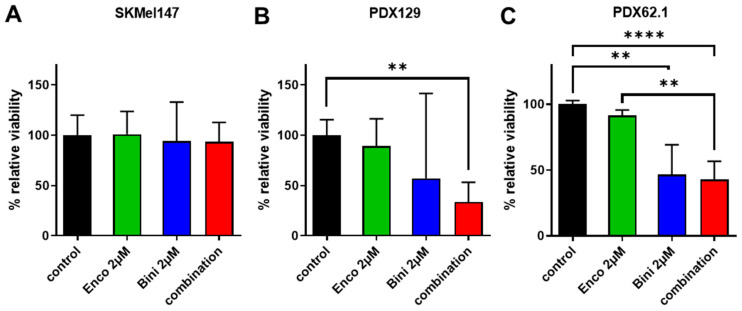
Effects of binimetinib and encorafenib on the ex vivo tumor viability of NRAS-mutated tumors. Slices (400 µm) of NRAS-mutated tumors were prepared using a vibratome and treated with binimetinib, encorafenib, or the combination. Alamar blue viability measurement of NRAS-mutated tumors (**A**) SKMel147, (**B**) PDX129, and (**C**) PDX62.1 treated with binimetinib, encorafenib, or the combination for 96 h. (**: *p* ≤ 0.01; ****: *p* ≤ 0.0001.)

**Figure 4 cancers-15-05521-f004:**
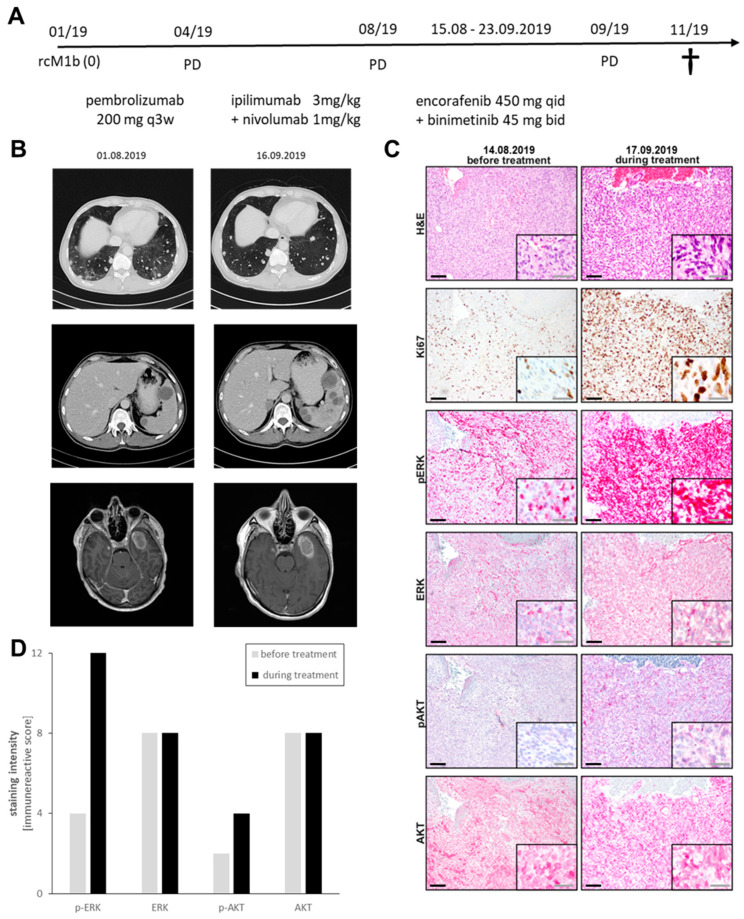
Effects of binimetinib and encorafenib in vivo in a patient with an NRAS-mutated tumor. (**A**) Treatment scheme for the patient. (**B**) CT scans of the patient before and during treatment. (**C**) IHC staining of the patient tumors before and during treatment for Ki67, pERK, ERK, pAKT, and AKT (black scales = 100 µm; grey scales = 50 µm). (**D**) Semiquantification of IHC staining. rcM1b = radiological and clinical M1b; q3w = every three weeks; qid = daily; bid = twice daily.

**Figure 5 cancers-15-05521-f005:**
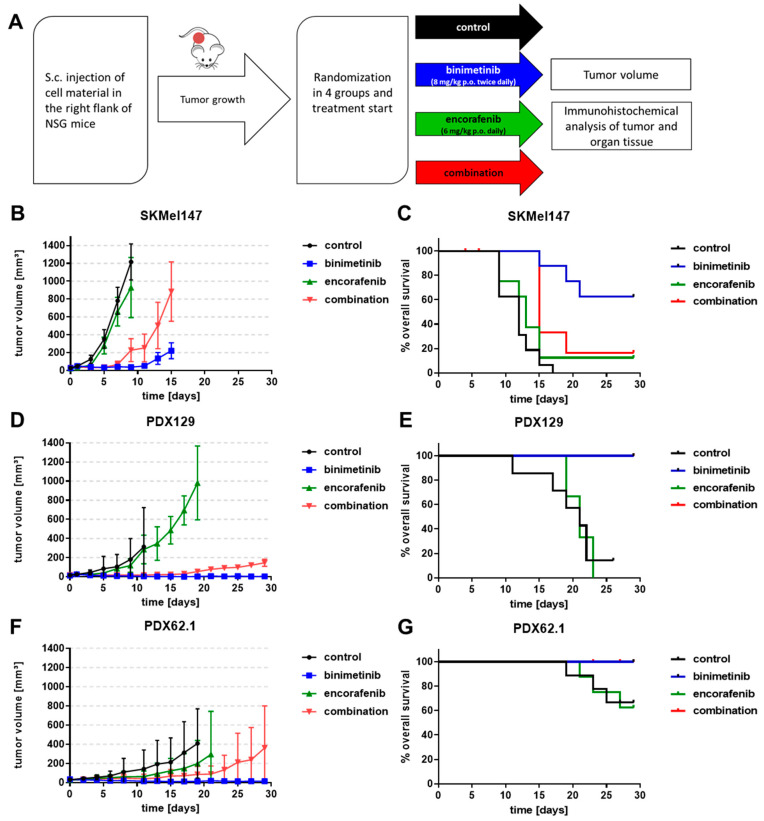
Effects of binimetinib and encorafenib in vivo in an NSG mouse model injected subcutaneously with melanoma cells. (**A**) Treatment scheme of the in vivo mouse experiment. (**B**,**D**,**F**) Measurement of the tumor volume of mice injected with SKMel147 cells (**B**), PDX129 cells (**D**), or PDX62.1 cells (**F**) during treatment with binimetinib (8 mg/kg p.o. twice daily), encorafenib (6 mg/kg p.o. daily), and the combination; the curves end when the first animal had to be terminated because of tumor volume. (**C**,**E**,**G**) Overall survival of mice subcutaneously injected with SKMel147 cells (**C**), PDX129 cells (**E**), or PDX62.1 cells (**G**) and treated with binimetinib (8 mg/kg p.o. twice daily), encorafenib (6 mg/kg p.o. daily), and the combination. Events in the overall survival plots are defined by the abort criteria for the mouse experiments (i.e., tumor diameter, ulceration, health score). The start of therapy is day 0. Red curves are sometimes hidden by blue curves. Censored events (marked by ticks) in the survival curves indicate animals which were removed from the experiment due to health issues but not due to tumor burden.

**Figure 6 cancers-15-05521-f006:**
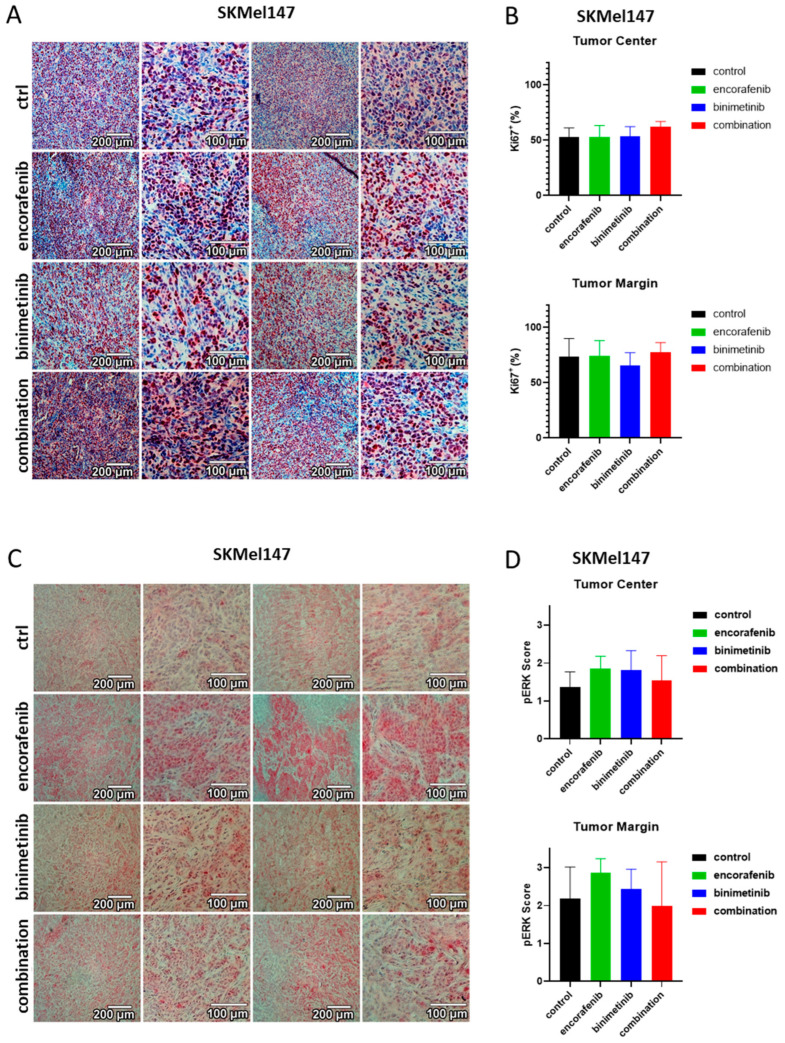
Effects of binimetinib and encorafenib on proliferation and pERK expression in vivo in an NSG mouse model injected subcutaneously with SKMel147 melanoma cells. (**A**) Ki67 staining of SKMel147 tumors under therapy with encorafenib, binimetinib, or the combination versus sham-treated control tumors. (**B**) Summary of the Ki67 semiquantifications of tumors from the SKMel147 melanoma model. Shown is the average fraction of proliferating cells in percent with standard deviations per therapy group. Asterisks indicate significant results (one-way ANOVA). (**C**) pERK staining of SKMel147 tumors treated with encorafenib, binimetinib, or the combination versus untreated control tumors. (**D**) Summary of the pERK semiquantifications of SKMel147 tumors. Shown is the average score of pERK staining intensities with standard deviations per therapy group. Asterisks indicate significant results (one-way ANOVA). Two examples at two magnifications are shown per condition.

**Figure 7 cancers-15-05521-f007:**
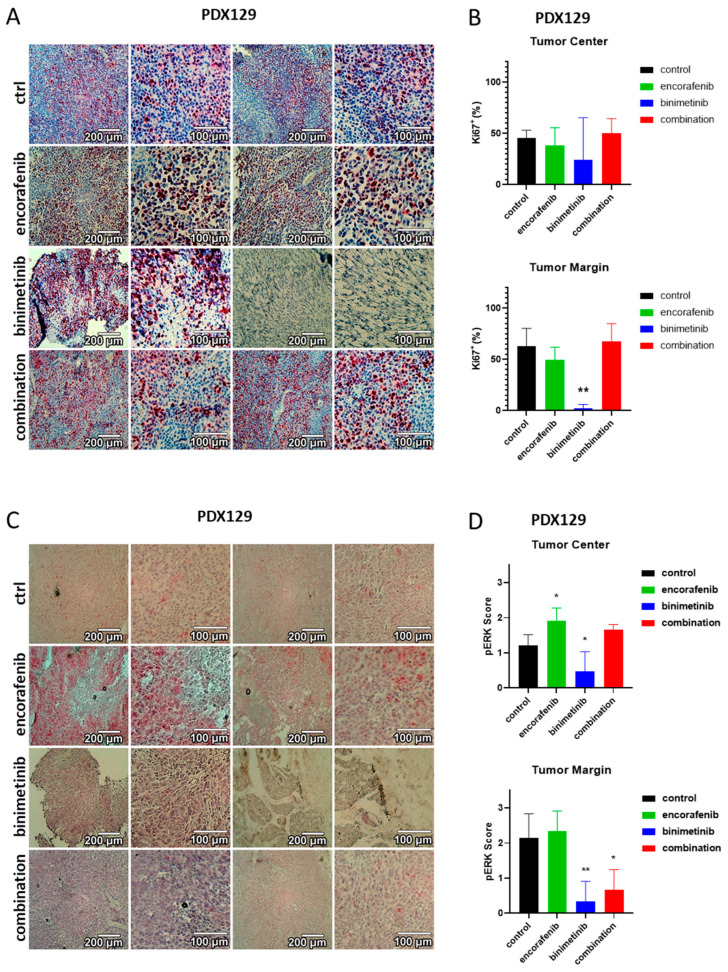
Effects of binimetinib and encorafenib on proliferation and pERK expression in vivo in an NSG mouse model injected subcutaneously with PDX129 melanoma cells. (**A**) Ki67 staining of PDX129 tumors under therapy with encorafenib, binimetinib, or the combination versus sham-treated control tumors. (**B**) Summary of the Ki67 semiquantifications of tumors from the PDX129 melanoma model. Shown is the average fraction of proliferating cells in percent with standard deviations per therapy group. Asterisks indicate significant results (one-way ANOVA). (**C**) pERK staining of PDX129 tumors treated with encorafenib, binimetinib, or the combination versus untreated control tumors. (**D**) Summary of the pERK semiquantifications of PDX129 tumors. Shown is the average score of pERK staining intensities with standard deviations per therapy group. Asterisks indicate significant results (one-way ANOVA). Two examples at two magnifications are shown per condition (*: *p* ≤ 0.05; **: *p* ≤ 0.01).

**Figure 8 cancers-15-05521-f008:**
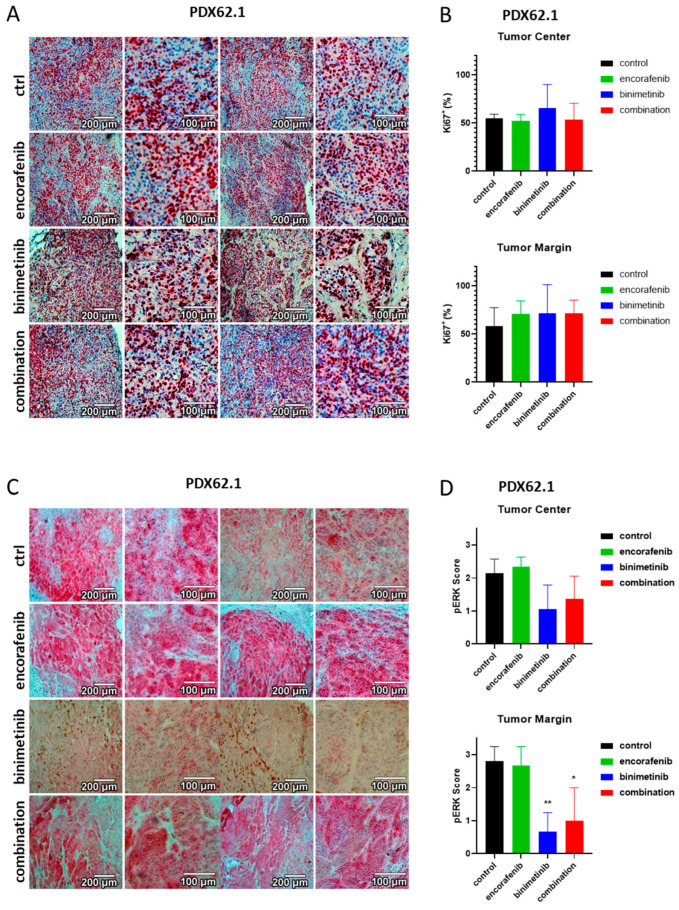
Effects of binimetinib and encorafenib on proliferation and pERK expression in vivo in an NSG mouse model injected subcutaneously with PDX62.1 melanoma cells. (**A**) Ki67 staining of PDX62.1 tumors under therapy with encorafenib, binimetinib, or the combination versus sham-treated control tumors. (**B**) Summary of the Ki67 semiquantifications of tumors from the PDX62.1 melanoma model. Shown is the average fraction of proliferating cells in percent with standard deviations per therapy group. Asterisks indicate significant results (one-way ANOVA). (**C**) pERK staining of PDX62.1 tumors treated with encorafenib, binimetinib, or the combination versus untreated control tumors. (**D**) Summary of the pERK semiquantifications of PDX62.1 tumors. Shown is the average score of pERK staining intensities with standard deviations per therapy group. Asterisks indicate significant results (one-way ANOVA). Two examples at two magnifications are shown per condition (*: *p* ≤ 0.05; **: *p* ≤ 0.01).

**Table 1 cancers-15-05521-t001:** Oligonucleotide primers used for quantitative reverse transcription PCR analysis.

Target	Oligonucleotide Primer 5′→3′	
*TBP*	for	tgc aca gga gcc aag agt gaa
	rev	cac atc aca gct ccc cac ca
*ATF4*	for	tgg gga aag ggg aag agg ttg taa
	rev	agt cgg gtt tgg ggg ctg aag
*CHOP*	for	aag gca ctg agc gta tca tgt
	rev	tga aga tac act tcc ttc ttg aac ac
*p8/NUPR1*	for	cca ttc cta cct cgg gcc tct catc
	rev	tct tgg tgc gac ctt tcc ggc

**Table 2 cancers-15-05521-t002:** Detailed information on PDX samples used in this study.

Sample	Mutational Status	Tumor Stage	Metastasis Site	Prior Treatment
PDX129	NRAS^Q61R^	IV	small intestine	Pembrolizumab; ipilimumab + nivolumab
PDX62.1	NRAS^Q61R^	IV	brain	Ipilimumab + nivolumab; DTIC

**Table 3 cancers-15-05521-t003:** Detailed information on IHC reagents used in this study.

Protein	Company	Retrieval	Dilution	Incubation	Detection
AKT	#4691 (Cell Signaling)	CC1 (#950-124 Ventana), 60 min	1:300	32 min, 37 °C	AP red (#760-501, Ventana)
pAKT	#4060 (Cell Signaling)	CC1 (#950-124 Ventana), 60 min	1:20	2 h, RT	AP red (#760-501, Ventana)
ERK	#9102 (Cell Signaling)	CC1 (#950-124 Ventana), 60 min	1:50	32 min, 37 °C	AP red (#760-501, Ventana)
pERK	#4376 (Cell Signaling)	CC1 (#950-124 Ventana), 60 min	1:200	2 h, RT	AP red (#760-501, Ventana)
Ki67	M7240 (Dako)	CC1 (#950-124 Ventana), 60 min	1:50	28 min, 37 °C	DAB brown uView (#760-700, Ventana)

**Table 4 cancers-15-05521-t004:** Definition of the immune reactive score (IRS).

% Positive Cells	Staining Intensity	IRS—% Positive Cells × Staining Intensity
0 = no positive cells	0 = no color reaction	0–1 = negative
1 = <10% positive cells	1 = mild reaction	2–3 = mild
2 = 10–50% positive cells	2 = moderate reaction	4–8 = moderate
3 = 51–80% positive cells	3 = intense reaction	9–12 = strongly positive
4 = >80% positive cells		

**Table 5 cancers-15-05521-t005:** IHC analyses of Ki67, (p)AKT, and (p)ERK in biopsies of a metastatic melanoma patient taken before and during treatment.

Marker	Before Treatment					During treatment				
	Intensity	% Positive Cells		IRS	Comments	Intensity	% Positive Cells		IRS	Comments
	Score		Score	Score		Score		Score	Score	
HE					Apoptosis, also necrosis					Little apoptosis, hardly any necrosis
MIB/Ki67		15–35% *		n.a.			30–60% *		n.a.	
p-ERK	2	50%	2	4	Heterogeneous	3	95%	4	12	
ERK	2	90%	4	8		2	90%	4	8	
p-AKT	1	10%	2	2		2	50%	2	4	
AKT	2	100%	4	8		2	100%	4	8	

* MIB1/Ki67 is routinely scored as % positive cells only, n.a. = not applicable.

## Data Availability

The data presented in this study are available in this article.

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
