# Peer review of "Exploring the In Vitro and In Vivo Therapeutic Potential of BRAF and MEK Inhibitor Combination in NRAS-Mutated Melanoma"

_cancers, 2023, doi:10.3390/cancers15235521_

Round 1
Reviewer 1 Report
Comments and Suggestions for Authors
Summary: This study investigates the therapeutic potential of combining BRAF and MEK inhibitors (BRAFi+MEKi) for the treatment of NRAS-mutated melanomas, which are known for their aggressive growth and limited treatment options. Using patient-derived xenografts (PDXs) and in vitro experiments, the researchers explore the efficacy and safety of this combination therapy. The study demonstrates that BRAFi+MEKi combination enhances apoptosis and inhibits tumor growth in NRAS mutant melanoma models. While in vivo results were not as pronounced as expected, binimetinib, a MEK inhibitor, still showed promise as a monotherapy. The findings offer valuable insights into potential precision treatments for NRAS mutant melanoma patients, advancing melanoma therapeutics.
Specific Comments:
· Please list the primer probes used for RT-PCR in Table 1 and the source manufacturer for oligonucleotide primer synthesis.
· Include any information available for patient tissue samples for section 2.8, like stage or prior treatments, as that would help understand the efficacy differences.
· Add cell culture conditions for Figure 2 & 3-exvivo tumor viability. Are the cells cultured in the presence of serum?
· Invitro culture of tumor slices for cell viability poses challenges with genetic drift and phenotyopic changes. Did the authors consider 3D exvivo assays to overcome these challenges. Including the details on the culture conditions may alleviate some of these concerns
Reference: Torsvik A, Stieber D, Enger PØ, Golebiewska A, Molven A, Svendsen A, Westermark B, Niclou SP, Olsen TK, Chekenya Enger M, Bjerkvig R. U-251 revisited: genetic drift and phenotypic consequences of long-term cultures of glioblastoma cells. Cancer Med. 2014 Aug;3(4):812-24. doi: 10.1002/cam4.219. Epub 2014 May 8. PMID: 24810477; PMCID: PMC4303149.
Strengths:
· The study addresses an important clinical problem - the limited treatment options for NRAS-mutated melanomas - and seeks to explore a potential solution.
· The research employs a combination of in vitro and in vivo models, including patient-derived xenografts, to comprehensively assess the therapeutic potential of the BRAFi+MEKi combination.
· The study provides mechanistic insights into how the combination therapy induces apoptosis and its potential impact on NRAS mutant melanomas.
Weakness:
· Limited In Vivo Combination Effect: The study reports that the in vivo data did not show a significant increase in combinatorial effects, which may raise questions about the clinical applicability of the combination therapy for NRAS mutant melanomas.
· Limited Methodology Details: The study lacks an in-depth description of the protocols and methodologies used in the experiments. Detailed protocols are essential for reproducibility and for other researchers to validate the findings and potentially apply the methods in their studies. A more comprehensive methods section would enhance the transparency and reliability of the research.

Comments on the Quality of English LanguageWell written
Author Response
We thank the reviewer very much for the constructive and useful comments. We think the additional information which is now integrated in the manuscript significantly improves the paper. In the following, we will discuss each of the reviewer's comments in detail.
Specific Comments:
- Please list the primer probes used for RT-PCR in Table 1 and the source manufacturer for oligonucleotide primer synthesis.
We thank the reviewer for this comment. We added the information that the primers were ordered from the company Biomers (https://www.biomers.net). Since we performed a real time PCR with sybr green and not a probe-based qPCR approach, the primers were not labeled.
- Include any information available for patient tissue samples for section 2.8, like stage or prior treatments, as that would help understand the efficacy differences.
We thank the reviewer for this comment and added the patient tissue information in the material and methods section 2.8 as Table 2.
- Add cell culture conditions for Figure 2 & 3-exvivo tumor viability. Are the cells cultured in the presence of serum?
We added the additional information to the material and methods sections 2.1 and 2.8. Yes, the medium contained 10% FCS.
- Invitro culture of tumor slices for cell viability poses challenges with genetic drift and phenotyopic changes. Did the authors consider 3D exvivo assays to overcome these challenges. Including the details on the culture conditions may alleviate some of these concerns
Reference: Torsvik A, Stieber D, Enger PØ, Golebiewska A, Molven A, Svendsen A, Westermark B, Niclou SP, Olsen TK, Chekenya Enger M, Bjerkvig R. U-251 revisited: genetic drift and phenotypic consequences of long-term cultures of glioblastoma cells. Cancer Med. 2014 Aug;3(4):812-24. doi: 10.1002/cam4.219. Epub 2014 May 8. PMID: 24810477; PMCID: PMC4303149.
We thank the reviewer for this comment and are well aware of this problem. Since we culture the slices for only 5 days after tumor resection, we assume that the problem of genetic drift and phenotypic changes will not occur in this short time.
Although we did not perform sequencing on the slices after this 5-day incubation period, we conducted functional comparisons between the tumor retrieved directly from the patient and the one passaged in the mouse. We found no changes in the morphology of the cells or their growth behavior (measured via Ki67 staining) in different PDX models.
Comparison of monolayer cell cultures with melanoma cells isolated directly from the patient tumor and those passaged in the mouse.
In the upper panel, cells isolated from different patient-derived tumors were stained for Ki67 (400x magnification). In the lower panel, the cells obtained from the same tumor were stained for Ki67 after passaging in NSG mice (400x magnification).
PLEASE SEE ATTACHMENT

Reviewer 2 Report
Comments and Suggestions for Authors
A study by Niessner et al., explores the therapeutic potential of encorafenib+binimetinib in NRAS-mutated melanoma, a genetic subtype of melanoma which has limited therapeutic options. A few in vitro and in vivo experimental models has been used.
Specific comments:
1. Gene names should be consistently written in italics. Editing of the text is required e.g, spaces behind [] for references etc.
2. Fig. 1 - please state in figure legend the time of incuvation with drugs. In addition, the choice of drug concentrations should be explained in text.
3. Fig. 2 and 3 - are any results statistically significant?
4. A general problem in using BRAFi in BRAF-wild type melanomas is associated with hyperactivation of ERK, which can be result in the development of non-melanoma skin cancers. Have the Authors tested the drug combination in normal skin cells (melanocytes, fibroblasts etc.)? In Fig. 2D it can be found that in cells exposed to drug combination p-ERK level is still higher than in control.
5. The origin of two PDX samples has not been described in Materials and Methods. This is particularly essential as these samples show generally different response pattern (e.g., in Fig. 3) than SkMel147 cells.
6. I recommend to put the paragraph 3.4 at the end of the result section. Fig. 4C lacks scale bars.
7. Discussion is poorly written, the Authors refer to only a few other papers. This must be revised.
Comments on the Quality of English Languageminor revision
Reviewer 3 Report
Comments and Suggestions for Authors
The manuscript by Niessner et al. describes a series of failed attempts to establish a synergy between a BRAF inhibitor encorafenib and a MEK inhibitor binimetinib against NRAS-mutant melanoma. After failing to produce conclusive evidence of cooperation in vitro, the authors proceed to a failed attempt at treating an NRAS-mutant clinical case, and then to a failed attempt at treating NRAS-mutant xenografts in mice. The manuscript is rife with problems, some of which are outlined below.
1. Statistical analysis is essentially lacking. The significances of differences, if any, is not indicated in the figures. Statistical analysis is especially important for this study because the claim of synergy requires a special type of analysis, and because the experiments where multiple conditions are evaluated require a correction for multiplicity of comparisons instead of a simple pairwise testing. Consequently, no credible conclusions can be drawn from most of the presented results. This lack of diligence means that the manuscript is unsuitable even as a report of negative findings.
2. Encorafenib inhibits its targets with EC50 in sub-nanomolar to single nanomolar range. The doses that show some effect, however small, in the presented experiments are orders of magnitude higher. This makes it very likely that this represents an off-target activity of the drug. Furthermore, no consideration is given to the possibility of steadily maintaining such a high concentration in patients.
3. The title of the manuscript is problematic. The use of “BRAFi” implies that the effects are typical to inhibitors of BRAF, which may be misleading. Despite the obvious concerns about the off-target activity of high-dose encorafenib, no attempt was made to prove that any purported effect is indeed due to BRAF inhibition. The report is also not limited to “in vivo study”, and the treatment of a human patient is hardly “preclinical”.
4. The statistical analysis is missing, but, apparently, there is no difference in the sub-G1 population between binimetinib alone or in a combination with encorafenib (Fig.1). This directly contradicts the claim that BRAFi significantly increases apoptosis induced by MEKi.
5. There are numerous problems with how the paper is written. For example, the sentence “BRAFi increased pERK..” is essentially duplicated in the “Methods” and “Results” parts of the abstract. The Materials and Methods section appears to be written for an entirely different study, as it describes the investigation of cooperation between vemurafenib and ascorbate (self-plagiarism from another paper). The data in Fig.3 are apparently referred to as Fig.2 in the text. Etc., etc.
6. The X-axis in Fig, 1 lacks an adequate label. It does not show the units, and the legend fails to explain whether the concentration refers to one or both drugs in the combination treatment.
7. While Fig.2 compares the drug –treated cells to those treated with DMSO, Fig.1, apparently, compares them to untreated cells. This is an important issue: the small purported effect in Fig.1 is seen only at very high doses, and, depending on the exact preparation of encorafenib, could easily come from DMSO, rather than the drug itself.
8. It is unclear why the data in Fig.3 is discussed in terms of the effect of the drug combination vs the untreated control. The real question is whether addition of encorafenib enhances the effect of binimetinib. And the answer is clearly negative. Plus, there is absolutely no effect of any treatment on SKMEL147 in Fig.3.
9. The authors have observed the well-known “paradoxical” oncogenic effect of BRAFi, which is the exact reason why such drugs are not used in NRAS-mutant cases. The claim that in the presented experiments “binimetinib was able to counteract this paradoxical activation of the MAPK pathway” is dubious, as the level of pERK in combination treatment are still higher than in the untreated control. That is, encorafenib effectively reverses the main anti-cancer effect of binimetinib.
10. The report of the clinical use of the drug combination is the most troubling part of the manuscript. Apparently, the combination was used in a patient after its inefficiency was already obvious from preclinical experiments. The danger of this combination was foreshadowed, for example, by the cumulative increase in pERK in Fig. 2. Predictably, the treatment enhanced the oncogenic signaling and the proliferation of cancer cells (as judged by Ki67). Therefore, it is likely that the patient was harmed by the treatment, and the harm was predictable from the preclinical data. This brings up important ethical issues. Perhaps, it might be reasonable to publish this observation as a case report in a specialized journal, as a caution to those who, for whatever reason, might contemplate a similar treatment.
11. Also predictably, encorafenib impeded the ability of binimetinib to suppress the growth of xenografts in mice. Once again, the only meaningful comparison is between the binimetinib- and the combination-treated arms, which strongly argues against the combination. The PDX62.1 study is especially confusing, as the tumor measurements for 3 of the 4 arms were stopped early (Fig.5F), while the survival of the same mice is shown for much longer (Fig.5G).
12. The Discussion section is inadequate. It appears that the authors try to downplay the failures of the proposed drug combination in all the settings and continue to advocate for its future use and exploration. Furthermore, if the ER stress is the only reason why encorafenib is included in the regimen, it is unclear why not to combine MEKi with much more specific proteotoxic agents, which, unlike BRAFi, are not expected to stimulate oncogenic pathways.

Reviewer 4 Report
Comments and Suggestions for Authors
The article provides a detailed investigation into the combination of BRAF and MEK inhibitors for the treatment of NRAS-mutated melanoma. The research is conducted in vitro and in preclinical in vivo models using patient-derived xenografts (PDXs). The article is well-structured and generally well-written, with clear objectives and findings. However, there are some areas that require improvement and clarification.
The abstract provides a concise summary of the study's objectives and key findings. It effectively highlights the significance of the research. Additionally, it would be beneficial to include some quantitative results or outcomes in the abstract, providing readers with a better understanding of the findings before delving into the full article.
The introduction effectively sets the stage by explaining the context and significance of the study, especially for patients with NRAS-mutant melanoma, but the references are quite old – the introduction needs updated (newer) references. Clinical trials need to be included here (https://trialsearch.who.int/)! It highlights the rationale for combining BRAF and MEK inhibitors based on previous research. However, it could benefit from a brief statement of the study's objectives and hypotheses to give the reader a clear roadmap for what to expect in the article.
The article explores a promising avenue for treating NRAS-mutated melanoma with detailed and quantifiable data presentation in the results section. Moreover, it includes comprehensive details on the experimental methods to ensure reproducibility.
Comments on the Quality of English LanguageConduct a final, thorough proofreading pass to catch any lingering grammatical or typographical errors.
Author Response
We thank the reviewer very much for the constructive and useful comments. We think that the additional references which are now integrated in the introduction of the manuscript significantly enhance the paper’s quality.
According to the reviewer’s suggestion we described the results in more detail in the abstract.
Round 2
Reviewer 2 Report
Comments and Suggestions for Authors
Thank you for very in-depth revision.
Author Response
Many thanks! We're delighted that our responses were able to meet the reviewer's satisfaction.
Reviewer 3 Report
Comments and Suggestions for Authors
The manuscript was significantly improved from the original version: it now gives a more accurate description of the failure of the drug combination. Also, several figures and figure legends were improved, and self-plagiarism in the Materials and Methods section has been corrected. However, some concerns remain.
1. The Conclusions section of the abstract still claims that “In vitro and ex vivo, a tendency towards a better response to the combination therapy was detected.” This is a misleading statement and should be removed. In most cases, the response to the combination was not better than to binimetinib alone. In fact, this statement now contradicts the Results section of the abstract, which now correctly lists the failures of encorafenib to improve on binimetinib alone.
2. There is still an absence of a statistical analysis for some experiments. Importantly, there is no statistical analysis for the cell cycle data in Fig. 1.
3. There are still problems with inadequately labeled and captioned figures. For example, there are no labels under the X-axis in Fig.1F and 1I. Also, contrary to what is stated in the rebuttal letter, Fig.1 legend still does not explicitly state that the numbers on the X-axes of the viability plots (D, G, J) refer to both drugs.
4. While in the rebuttal letter the authors agreed that the need for very high concentrations of encorafenib probably indicates the reliance on an off-target activity of the drug, they still fail to mention this critically important point in the Discussion.
5. In the rebuttal letter, the authors reasonably discuss the possibility to avoid the BRAFi “paradox” in NRAS-mutant melanomas either by using inhibitors with different modes of action (e.g. pan-RAF inhibitors) or by targeting ER stress directly. However, the Outlook section still claims that “in vitro data suggest that there may be a therapeutic window in which this paradoxical activation can be avoided”, seemingly referring to the use of encorafenib in the manuscript. The only actual data on in vitro pathway activation in this manuscript (Fig. 2D) does not support this statement: it shows increased pERK upon addition of encorafenib. It would be appropriate to remove this statement, possibly replacing it with the ideas on the future synergistic combinations from the rebuttal letter.
Author Response
- The Conclusions section of the abstract still claims that “In vitro and ex vivo, a tendency towards a better response to the combination therapy was detected.” This is a misleading statement and should be removed. In most cases, the response to the combination was not better than to binimetinib alone. In fact, this statement now contradicts the Results section of the abstract, which now correctly lists the failures of encorafenib to improve on binimetinib alone.
- As the reviewer suggested, we have replaced this sentence with the following: “In in vitro and ex vivo settings, the combination therapy was observed to elicit a response; however, it did not amplify the efficacy observed with binimetinib alone.
- There is still an absence of a statistical analysis for some experiments. Importantly, there is no statistical analysis for the cell cycle data in Fig. 1.
- We thank the reviewer for this comment and added statistical analysis now also to the cell cycle data in figure 1.
- There are still problems with inadequately labeled and captioned figures. For example, there are no labels under the X-axis in Fig.1F and 1I. Also, contrary to what is stated in the rebuttal letter, Fig.1 legend still does not explicitly state that the numbers on the X-axes of the viability plots (D, G, J) refer to both drugs.
- We thank the reviewer for this comment and added the missing labeling. Also, we added to the figure legend for 1D, 1G and 1J: Both drugs were used in equimolar ranges, so the x-axis labeling applies to both drugs.
- While in the rebuttal letter the authors agreed that the need for very high concentrations of encorafenib probably indicates the reliance on an off-target activity of the drug, they still fail to mention this critically important point in the Discussion.
- We apologize for the inadequate mention of the role of the putative off-targets of encorafenib in the discussion of our manuscript and have made up for this accordingly. First, we mentioned that the paradoxical hyperactivation of the MAPK pathway can be interpreted as an off-target effect (line 589-590: “This can be interpreted as an off-target effect of the BRAFi”). Secondly, we added the information that further off-targets might be the cause for ER stress induction (line 610-613: “We assume that higher levels of BRAFi are required to induce ER stress, and this may be the result of off-target effects that also affect the endoplasmic reticulum although no off-targets other than RAF isoforms are known for encorafenib”.
- In the rebuttal letter, the authors reasonably discuss the possibility to avoid the BRAFi “paradox” in NRAS-mutant melanomas either by using inhibitors with different modes of action (e.g. pan-RAF inhibitors) or by targeting ER stress directly. However, the Outlook section still claims that “in vitro data suggest that there may be a therapeutic window in which this paradoxical activation can be avoided”, seemingly referring to the use of encorafenib in the manuscript. The only actual data on in vitro pathway activation in this manuscript (Fig. 2D) does not support this statement: it shows increased pERK upon addition of encorafenib. It would be appropriate to remove this statement, possibly replacing it with the ideas on the future synergistic combinations from the rebuttal letter.
We agree with the reviewer and have followed his advice. Specifically, we have deleted the sentence referred to in bullet point 5 and replaced it with the following sentence: “However, in vitro data suggest that combinations such as panRAFi and MEKi can induce ER stress, causing melanoma cells to initiate programmed cell death. As this effect has yet to be confirmed in vivo, further research on such novel drug combinations is needed to develop an effective therapy against NRAS-mutated melanomas.”